# Structure and Mechanical Properties of Multi-Walled Carbon Nanotubes-Filled Isotactic Polypropylene Composites Treated by Pressurization at Different Rates

**DOI:** 10.3390/polym11081294

**Published:** 2019-08-02

**Authors:** Xiaoting Li, Wenxia Jia, Beibei Dong, Huan Yuan, Fengmei Su, Zhen Wang, Yaming Wang, Chuntai Liu, Changyu Shen, Chunguang Shao

**Affiliations:** Key Laboratory of Materials Processing and Mold (Zhengzhou University), Ministry of Education, National Engineering Research Center for Advanced Polymer Processing Technology, Zhengzhou University, Zhengzhou 450002, China

**Keywords:** isotactic polypropylene, multi-walled carbon nanotubes, pressurization rate, crystalline behavior, mechanical properties

## Abstract

Isotactic polypropylene filled with 1 wt.% multi-walled carbon nanotubes (iPP/MWCNTs) were prepared, and their crystallization behavior induced by pressurizing to 2.0 GPa with adjustable rates from 2.5 to 1.3 × 10^4^ MPa/s was studied. The obtained samples were characterized by combining wide angle X-ray diffraction, small angle X-ray scattering, differential scanning calorimetry, transmission electron microscopy and atomic force microscopy techniques. It was found that pressurization is a simple way to prepare iPP/MWCNTs composites in mesophase, γ-phase, or their blends. Two threshold pressurization rates marked as **R_1_** and **R_2_** were identified, while **R_1_** corresponds to the onset of mesomorphic iPP formation. When the pressurization rate is lower than **R_1_** only γ-phase generates, with its increasing mesophase begins to generate and coexist with γ-phase, and if it exceeds **R_2_** only mesophase can generate. When iPP/MWCNTs crystallized in γ-phase, compared with the neat iPP, the existence of MWCNTs can promote the nucleation of γ-phase, leading to the formation of γ-crystal with thicker lamellae. If iPP/MWCNTs solidified in mesophase, MWCNTs can decrease the growth rate of the nodular structure, leading to the formation of mesophase with smaller nodular domains (about 9.4 nm). Mechanical tests reveal that, γ-iPP/MWCNTs composites prepared by slow pressurization display high Young’s modulus, high yield strength and high elongation at break, and meso-iPP/MWCNTs samples have excellent deformability because of the existence of nodular morphology. In this sense, the pressurization method is proved to be an efficient approach to regulate the crystalline structure and the properties of iPP/MWCNTs composites.

## 1. Introduction

The reinforcement of polymers with nano-scaled fillers have attracted great attention for the past ten years, and it has been widely accepted that this reinforcement can improve mechanical, and other properties, including changes in polymer crystallization behavior [1,2,3,4]. Of the potential fillers for polymer composites, the carbon nanotubes (CNTs) has taken increased interest mainly due to their exceptional mechanical properties, outstanding thermal conductivity and electrical properties [5,6,7].

Among a variety of polymers filled with CNTs, isotactic polypropylene (iPP) is one of the most preferred thermoplastics that is commonly reinforced with CNTs [8,9,10]. It has been reported that mechanical properties of iPP such as tensile strength, modulus, and fracture toughness can increase dramatically at very low CNTs contents (<1%) [4,11], owing to the large nanocarbon/polymer interface arising from their high surface-to-volume ratio. Furthermore, the crystallization behavior of iPP matrix can also be greatly affected by CNTs, which may improve performances of iPP or to achieve new properties of the composites. Grady et al. firstly reported that the crystallization rate was substantially higher in the filled iPP than the unfilled iPP under isothermal or nonisothermal crystallization process [12]. Xu and Wang showed that the formation of the CNTs network could mainly restrict the mobility and diffusion of iPP chains to crystal growth fronts [13]. Typically, CNTs are considered to be nucleation agents that accelerate iPP crystallization, while CNTs networks can impose physical confinement on polymer crystal growth. Besides, CNTs can also induce appearance of a new crystal structure. For instance, Grady et al. noted that CNTs can promote growth of β-form iPP at the expense of α form under isothermal crystallization process, and Zhang et al. reported that when iPP is infiltrated into nanotube aerogel fibers, both α-phase and γ-phase can be formed and γ-phase is the major component caused by the geometric confinement [14]. Interestingly, Ning et al. reported that CNTs bundles can promote iPP chain alignment along the CNT axis, possible to induce the formation of a iPP ‘nanohybrid shish kebabs’ structure [15]. In short, addition of CNTs can not only affect the crystallization kinetics of iPP, but also induce new crystal structures as well as potentially improve the iPP performance.

In numerous research on iPP/CNTs, the common theme focused on was their crystallization behavior under atmospheric pressure [16,17], but less attention was paid to their solidification behavior induced by high pressure, neither is there any report on the effect of pressurization rate on melt solidification. More recently, several studies proved that pressurization is an efficient approach to prepare polymer materials with special performance, such as living polymer sulfur which exhibits exceptional thermodynamic and kinetic stability, glassy polyether-ether-ketone samples possessing excellent friction and considerable stiffness, and the glassy poly (lactic acid) which shows better cold crystallization performance [18,19,20]. Additionally, our recent work indicated the crystal structure and morphology of iPP can be accurately controlled by adjusting pressurization conditions [21], and firstly proved high pressure-annealing can induce meso-γ transformation [22]. These results above have confirmed that pressurization is a novel method to tailor the crystalline structure of iPP. Besides, pressurization process is superior to temperature cooling preparation, because stress equilibrates within the polymer melts faster than thermal equilibration, thus the pressurization treatments can get rid of thermal conductivity of the melts and would produce products with more uniform structure with larger size, and this is especially common for polymers which have low thermal diffusivities [18]. More importantly, pressurization conditions can be precisely controlled using conventional techniques, which has potential values for industrial applications.

It is known that high pressure is beneficial for γ-iPP formation in the pressure-crystallized iPP [23]. γ-crystal is formed by bilayers and the direction of the chain axis in adjacent bilayers is tilted by approximately 80° against each other, which is a unique packing arrangement. γ-spherulites do not show the cross-hatching feature characteristic of α-spherulites, but show only a moderate branching with lamellae packed in parallel stacks and spread radially [24]. Because of the bilayers texture and/or immature lamellar structure [25], γ-form iPP shows better mechanical performance than α-form iPP. In this study, employing a home-made pressure device, we prepared iPP/multi-walled carbon nanotubes (MWCNTs) composites in mesophase, γ-phase and their blends through adjusting the pressurization rate in a broad range, and for the first time investigated the composites crystallization behavior during that pressurization. Furthermore, we studied the influence of crystalline structure and MWCNTs on mechanical properties of the prepared composites.

## 2. Materials and Methods

### 2.1. Materials

The iPP material (grade T30s) was obtained from Dushanzi Petroleum Chemical Co., Xinjiang, China, which had an average molecular weight Mw = 399 kg/mol and a polydispersity Mw/Mn = 4.6. MWCNTs of outer diameter (o.d.) 50–70 nm and length 10–20 μm were purchased from Chengdu Organic Chemicals Co. Ltd., Chengdu, China, the Chinese Academy of Sciences R&D Center for Carbon Nanotubes.

### 2.2. Preparation of iPP/MWCNTs Composites

The iPP granules and MWCNTs were firstly dried in a vacuum oven at 80 °C for 8 h, then a masterbatch of iPP and MWCNTs with 5 wt.% MWCNTs was produced by melt mixing in a twin-screw extruder (Haake MiniLab II, Breda, The Netherlands) for 10 min at 30 rpm and 200 °C. Afterwards, nanocomposites with 1 wt.% MWCNTs were produced by diluting the masterbatch under the same processing condition. Then raw iPP granules and the composites were compression-molded into the circular samples with the diameter of 24 mm and the thickness of 1 mm respectively.

The above circular samples were treated by compression with a home-made high pressure device shown in Figure 1a, whose detailed information has been described in previous work [21,22]. The compression treatment consists of four steps as given in Figure 1b: First, the samples were pre-pressurized at 10 MPa and heated to 200 °C, then annealed isothermally for 10 min to erase the processing histories. Second, the melting samples at 200 °C were pressurized to 2.0 GPa within the controllable time of 0.15, 20, 25, 40, 60, 80, 100, 200, 400, 800 s respectively, which correspond to different pressurization rates. Third, the pressurized samples were immediately cooled down to 40 °C at an average cooling rate of 10 °C/min under 2.0 GPa. At last, the pressure was released in 5 s. For comparison, we also prepared the quenching samples (by immersing the melts into ice water) and the natural cooling samples.

### 2.3. Tensile Experiments

Mechanical tests were performed using a testing instrument (linkam TST-350, Linkam Scientific Instruments Ltd., Tadworth, UK) under the controlled experimental room temperature (25 °C). All samples were cut into dumbbell-shape of 21.0 mm in total length, 3.0 mm in neck length and 2.0 mm in neck width. The samples were mounted between two clamps and a constant extension rate of 0.6 mm/min was applied. The deformation was calculated as:(1)$$\varepsilon =[(L_{f}-L_{0})/L_{0}]\times 100$$
where $L_{0}$ and $L_{f}$ are the initial and final specimen lengths respectively. The stress is calculated through dividing the tensile force F by the initial cross-section area $A_{0}$ as follows:(2)$$\sigma =F/A_{0}$$

For all the mechanical tests, five samples were repeatedly-tested and the standard deviations were calculated.

### 2.4. Characterizations

#### 2.4.1. Wide-Angle and Small-Angle X-ray Measurements

Wide-angle X-ray Diffraction measurements were performed using X-ray diffraction (WXRD, D8 discovery system, Bruker, Karlsruhe, Germany). The X-ray wavelength was 1.54 Å and the sample-to-detector distance was 85.0 mm. The WAXD patterns were recorded by a vant 500 two-dimensional (2D) detector. Each pattern was acquired with the 2θ ranging from 5 to 32° and the exposure time was 100 s. Small-angle X-ray scattering measurements (SAXS, Nanostar, Bruker, Germany) were conducted, and the sample-to-detector distance was fixed as 2508 mm. SAXS images were recorded with a Pilatus 100 K detector of Dectris, Baden, Swiss, and the exposure time was 300 s. 2D SAXS and WAXD patterns were conversed by Fit2D software into one-dimensional data (1D). The 1D SAXS data were Lorentz-corrected by multiplying the intensity value by q^2^. The 1D WAXD curves were fitted according to Gaussian functions to obtain the crystallinities.

The phase content can be estimated by the following equation based on an interactive peak-fit-procedure [26]:(3)$$X_{meso}=A_{meso}/\left(A_{meso}+A_{\gamma }+A_{amorp}\right)$$
(4)$$X_{\gamma }=A_{\gamma }/\left(A_{meso}+A_{\gamma }+A_{amorp}\right)$$
where, $A_{meso}$, $A_{\gamma }$ and $A_{amorp}$ are the fitted areas of mesophase, γ-phase and amorphous region, respectively.

#### 2.4.2. Differential Scanning Calorimeter

A differential scanning calorimeter (DSC-Q2000, TA, New Castle, DE, USA) was employed using nitrogen as protective gas. The temperature and heat flow were calibrated by means of indium according to a standard program. Samples of about 4 mg were selected from the pressurized samples and inserted into small aluminum pans. The trace of DSC scans was recorded upon heating from 30 to 200 °C at rate of 10 °C/min.

#### 2.4.3. Transmission Electron Microscopy (TEM)

TEM measurements (JEM-1230, JEOL, Tokyo, Japan) were conducted on iPP/MWCNTs composites. Before the measurement, the samples were in advance cut at −160 °C into thin films of about 100 nm using the ultramicrotome (EM FC7, Leica, Vienna, Austria).

#### 2.4.4. Atomic Force Microscope

Atomic force microscope measurements (AFM, MultiMode 8, Bruker, Germany) were performed with a Nanoscope V controller. The samples were cut at −120 °C using the ultramicrotome (EM FC7, Leica, Austria) to obtain a flat surface.

## 3. Results and Discussion

### 3.1. Pressurization-Induced Formation of Meso-iPP/MWCNTs Composites

Structures of the iPP/MWCNTs composites prepared by ultra-high pressures of 2.0 GPa with a relatively high pressurization rate of 100 MPa/s (under this condition the obtained samples were defined as RC samples) were characterized by WAXD as shown in Figure 2a and compared with that of the quenching composites. According to the WAXD profiles, no crystalline reflection peak was observed in RC sample, but two broad scattering halos corresponded to mesomorphic iPP were found at 2θ of 15.0° and 21.0° respectively [27]. The inserted 2D-WAXD pattern displays that the mesomorphic composites obtained by pressurizing was not oriented, meaning the mould utilized in the pressurization process was well sealed and the melt did not flow [28]. For the quenching samples, the characteristic diffraction peaks of (110), (040), (130), and (111) crystallographic planes of α-iPP were identified, demonstrating that the cooling rate of the ice water quenching is not fast enough to inhibit the crystallization in iPP/MWCNTs composites [29]. These results proved again that pressurization is an effective approach to prepare the metastable state, which seems superior to temperature quenching [30]. The AFM phase image of the mesomorphic composites were displayed in Figure 2b, which demonstrates that these samples have distinct nodular structure as quenching-mesophase, while the average size of the nodular domains in mesomorphic composites is about 9.0 nm, slightly smaller than that of the quenching-mesophase (about 10–20 nm) [31].

It was shown that the CNTs can be efficiently dispersed into the polymer matrix, and the composites displayed a homogeneous microstructure when the CNTs content was low (<1%) [5]. Representative TEM images of RC samples were given in Figure 3a,b, which shows the MWCNTs were basically uniform-dispersed in the iPP matrix and few agglomerates remained. In higher magnification, it was easy to find that some MWCNTs remained bent and even interwoven in the iPP matrix. This kind of bending or knot may significantly reduce the reinforcement structure provided by MWCNTs to polymer matrix compared with the theoretical reinforcement provided by direct inclusion [32].

### 3.2. Effect of Pressurization Rate on Polymorphism of Crystallization

iPP/MWCNTs composites after high pressure treatment are characterized by WAXD. Figure 4a shows the selected WAXD profiles of the composites prepared at different pressurization rates, and the related representative 2D-WAXD patterns were given in Figure 4b. It shows that no orientation of the mesophase and/or crystalline phase was detected for the all samples, meaning that no flow occurred during the pressurization process [28]. Clearly, the modification of crystallites formed exhibits a strong dependence on pressurization rate. For the low pressurization rate at 2.5 MPa/s, the characteristic diffraction peaks of (111), (008), (117), and (202) crystallographic planes were identified, demonstrating that the γ-phase was generated. With increasing of the pressurization rate, the amount of formed γ-phase decreased significantly. When the pressurization rate reached 100 MPa/s or higher, the formation of γ-phase was completely suppressed and pure mesophase was obtained.

Figure 4c shows a representative fitting performed on the WAXD profile, where multiphases coexist. WAXD profiles were analyzed by a deconvolution procedure [33], in which the full spectrum was considered as a superposition of a number of reflections. During the peaks fitting process, 7 reflections were considered: 4 for γ-phase, 2θ = 13.9° (111), 16.7° (008), 20.1° (117) and 21.5° (a superposition peak); 2 for mesophase, corresponding to 2θ = 15.0° and 21.0°; 1 for amorphous, 2θ = 17.0°; each reflection was described by a combination of a Gaussian function. Figure 4d shows the quantitative dependence of mesophase, amorphous and γ phases on pressurization rate, and three regions (separated by dotted lines) were clearly identified. When the pressurization rate was below 5.0 MPa/s, only γ-phase forms, and the fractions of γ-phase and amorphous phase were 45% and 55% respectively. Once the pressurization rate exceeded 5.0 MPa/s, the amount of γ-phase began to decrease but that of mesophase increased, thus the rate of 5.0 MPa/s was denoted as **R_1_** which determines whether the mesophase can generate. When the pressurization rate reached 100 MPa/s (denoted as **R_2_**) or higher, pure mesophase could be obtained, and the fractions of mesophase and amorphous increased to 32% and 68% respectively.

It is known that MWCNTs can promote crystallizing of iPP melt into α-iPP during the cooling or even ice quenching process (Figure 2a), since MWCNTs mainly functioned as nucleating agents for the α-phase and could change the crystallization rate or even induce changes in crystal conformation [4,34]. To make clear the promotion effect of MWCNTs on the crystallization behavior of iPP melts under pressurization process, the crystallization evolution of neat iPP melts under pressurization treatment were investigated and compared. Figure 5a shows the selected WAXD profiles of neat iPP under 2.0 GPa at different pressurization rates, and Figure 5b summarizes the changes of the contents of different phases. For neat iPP, two threshold pressurization rates were also identified, while the first one was about 2.5 MPa/s and the second one was about 10 MPa/s, which are both lower than that of the iPP composites. These shifts of the threshold pressurization rates could be explained in view of γ-phase crystallization ability. MWCNTs can promote γ-phase formation in the composited iPP and thus suppress mesophase formation, so that the pressurization rate needed for the mesophase formation (**R_1_**) and γ-phase disappearance (**R_2_**) are both higher than that of the neat iPP. Actually, Zhang and Bucknall et al. have proved that individual CNTs preferably nucleate γ-form iPP and even can induce growth of γ-form transcrystals on the condition that the content of CNTs was higher than 30 wt.%, and they pointed out geometric confinements leading to the formation of γ-iPP [14]. As mentioned above, γ-phase is the stable phase under high pressure, considering the heterogeneous nucleation ability of CNTs, the formation of γ-phase tends to be easier and faster in iPP/MWCNTs composites. As shown in Figure 5b, under the same pressurization rate the γ-phase content of the composites is remarkably higher than that of the neat iPP (Figure 5b).

Though it is confirmed that the formation of the γ-phase is preferred at high pressures and low degrees of supercooling, our results demonstrated that high pressurization rate promotes the formation of mesophase rather than the stable γ-phase for iPP/MWCNTs system. This phenomenon maybe obey the “law of successive states” formulated by Ostwald [35], which suggests that the new phase appears to go through stepwise changes from less to more stable polymorphs, when leaving a given state and transforming to another, the state to be sought out is not the thermodynamically stable one but the one nearest in stability to the original state [16]. Obviously, the γ-phase in iPP with special crystalline structure of molecular chain structure, which has the crossed bilayer with an angle of 80°, could not be the nearest state to the initial melt iPP. On the contrary, the mesomorphic iPP whose molecular chain conformation is an approximately parallel arrangement with lose three-dimensional long-range order is closer to the melt structure, so that it will be sought out due to the kinetic reason [33,36]. Additionally, Androsch and Wunderlich pointed out that the formation of mesophase is induced by the changes of the nucleation behavior of the melt, from heterogeneous nucleation at low supercooling to homogeneous nucleation at high supercooling [37]. These interpretations maybe suitable to explain the dependence of the mesophase formation on the pressurization rate in iPP/MWCNTs composites. According to the rapid increasing of the equilibrium melting point of iPP with pressure (about 300 °C GPa^−1^) [30,38,39], high pressurization rate can lead to high supercooling and thus change the nucleation behavior of the melt from heterogeneous nucleation to homogeneous nucleation, leading to the formation of mesophase.

### 3.3. Effect of Pressurization Rate on Nanoscale Structure of the Composites

Figure 6a shows 2D-SAXS patterns of iPP/MWCNTs composites prepared under different pressurization rate, and the relative information of neat iPP was also given for comparison. The Lorentz-corrected intensity was plotted as a function of scattering vector (q = 4π sin θ/λ) as shown in Figure 6b,c. For the composites treated at rate of 2.5 MPa/s, an obvious scattering peak at $q_{\max}$ = 0.61 nm^−1^ was detected, and it does not show obvious change until the pressurization rate reaches **R_1_** of 5 MPa/s. When that rate exceeds **R_1_** where the formation of mesophase begins, with its increasing $q_{\max}$ shifts to higher value and $I_{\max}$ remarkably decreases, meaning the changing of $q_{\max}$ and $I_{\max}$ is induced by the formation of mesophase. Furthermore, both $q_{\max}$ and $I_{\max}$ tend to be stable since the pressurization rate reaches **R_2_**, in which case complete mesophase will be obtained (Figure 4b). For the neat iPP, when the pressurization rate is 1.8 MPa/s, a scattering peak was detected at $q_{\max}$ = 0.73 nm^−1^, much higher than that of the composites, and it does not show obvious change until that rate exceeds 2.5 MPa/s, then $q_{\max}$ starts to shift to lower value and $I_{\max}$ simultaneously decrease, while both $q_{\max}$ and $I_{\max}$ become stable after it exceeds 10 MPa/s.

The scale of L can reflect the average distance between either γ lamellar domains or mesophase nodular domains, and it can be determined from the position of $q_{\max}$ based on the Bragg law:(5)$$L=2\pi /q_{\max}$$

Figure 7 shows changes of L as a function of pressurization rate. When the pressurization rate is lower than **R_1_**, L of the composites is about 10.8 nm, larger than that of neat iPP (about 9.5 nm), and this can be attributed to the formation of γ-crystal with thicker lamellae [40]. For the composites, the value of L keeps constant before the formation of mesophase, then L begins to decrease with the appearance of mesophase and reaches a smaller steady value of about 9.4 nm when the pressurization rate surpassed **R_2_**. Conversely, for neat iPP, the value of L starts to increase when the mesophase began to form, and continues increasing with the rising pressurization rate and finally keeps constant of about 10.5 nm. The results above show the average size of the nodular domains of meso-iPP/MWCNTs composite is smaller than that of neat meso-iPP. Actually, the existence of MWCNTs can increase the energy barrier for transport of iPP chains and thereby increase the viscosity of iPP matrix, thus the growth rate of mesomorphic iPP in the composites should be lower than that of neat iPP, which is to be responsible for the formation of smaller nodular domains [31].

Based on the above WAXD and SAXS results, three conclusions can be drawn. First, for composites, two threshold pressurization rates marked as **R_1_** and **R_2_** were identified. Second, during pressurization, MWCNTs can promote the crystallization of γ-iPP, therefore, under the same pressurization rate between **R_1_** and **R_2_**, the γ-phase content of the composites is greatly higher than that of the neat iPP. Third, the meso-iPP/MWCNTs composites also have distinct nodular structure as the neat mesophase, and their nodular average size are smaller than that of neat meso-iPP [21].

### 3.4. Thermal Preformance of the iPP/MWCNTs Nanocomposites

Figure 8 shows the DSC thermograms of neat meso-iPP and composited meso-iPP obtained under the pressurization rate of 1.3 × 10^4^ MPa/s, and the inset shows the melting peak region. It is clear to see that the thermal behaviors of these two samples are very similar. With the increasing of the temperature, they both exhibit a small exotherm at about 95 °C, which is associated with the recrystallization of the mesophase into α-phase during heating under atmospheric pressure. It is noteworthy that, if the monomorphic matrix melts in advance the MWCNTs can promote the meso-α transition. However, the recrystallization peaks of the neat and composited meso-iPP are similar, indicating that the carbon tubes had no effect on the meso-α transition, which proved that the mesophase transit directly to α-phase in the heating process instead of melting beforehand [41]. The endothermic peaks at around 160 °C were attributed to melting of the α-phase that recrystallized from the mesophase. The same melting peaks indicated that there is almost no difference in the perfection of the recrystallized α-crystals. In short, though the nodular domains average size of the composited meso-iPP are smaller than that of neat meso-iPP, it has almost no effect on the meso-α transition and the perfection of recrystallized α-crystals.

Figure 9a shows the DSC heating curves of the iPP/MWCNTs samples prepared by pressurizing to 2.0 GPa at different pressurization rates, in which the inset displays the melting peak region. When the pressurization rate is higher than 100 MPa/s, a standard DSC curves of mesomorphic iPP is obtained. When the pressurization rate decreases to 80 MPa/s or lower, the exothermic peak attributed to the meso-α transition starts to decrease, meaning the content of mesophase decreases. Once that rate decreases to 33.3 MPa/s, the melting peak of γ-phase located at about 155.0 °C becomes clear, and the melting temperature of γ-phase increased gradually with the rate further decreasing, indicating the perfection of γ-phase increased. When the pressurization rate is below 5 MPa/s, where only γ-phase can be detected by WAXD (Figure 5), the melting temperature of γ-phase increased to about 158.5 °C and keeps stable despite further decreasing of the pressurization rate.

To clarify the effect of pressurization rate on the perfection of γ-crystal, changes of melting temperatures of γ-phase in composited iPP are illustrated in Figure 9b. Obviously, the melting point of γ-phase reduces from 158.5 to 155.2 °C when the pressurization rate rises from 2.5 to 33.3 MP/s.

Lamellar thickness can be calculated based on its relationship with the melting point by Gibbs-Thomson equation [42]:(6)$$l=\frac{2\sigma_{s}T_{m}^{0}}{\Delta H(T_{m}^{0}-T_{m})}$$
where $T_{m}$ is the melting point, $T_{m}{}^{0}$ is the equilibrium melting point of γ-phase of about 187.2 °C at atmospheric pressure. The heat of fusion (ΔH) and the fold surface energies ($\sigma_{s}$) of γ-phase are determined to be 150.0 J/s and 53.6 erg/cm^2^ respectively [43]. The variation of the lamellar thickness for γ-iPP/MWCNTs composites as a function of pressurization rate is shown in Figure 9b, which shows that lamellar thickness decreases slightly from 4.6 to 4.2 nm as the pressurization rate rises from 2.5 to 33.3 MPa/s. Comparing with the crystallinity information provided by WAXD (Figure 4), the change of pressurization rate seems to have a greater influence on crystallinity than on the lamellar thickness of γ-phase. Additionally, under the same pressurization rate (for example 2.5 MPa/s), where only γ-iPP formed, lamellar thickness of neat γ-iPP is about 4.1 nm, a little smaller than that of composited γ-iPP (4.6 nm), which conforms to the SAXS results (Figure 7).

### 3.5. Mechanical Properties of the iPP/MWCNTs Nanocomposites

Stress-strain curves of the samples in γ-phase (crystallized from slow pressurization), mesophase (crystallized from rapid pressurization) and α-phase (crystallized from natural cooling) are summarized in Figure 10, which shows that all samples deform uniformly up to a critical strain beyond which they exhibit neck formation. For α-iPP/MWCNTs composites, samples exhibit highest modulus and strength but lowest elongation at break [11,44]. γ-iPP/MWCNTs was found having comparable modulus and strength as α-iPP/MWCNTs, consistent with the results by Lezak et al., where γ-iPP had comparable modulus and yield stress as α-iPP [25]. For meso-iPP/MWCNTs composites, the yield peak is broader and the yield stress is significantly weaker compared with the other samples. Moreover, the stress-strain curves also proved that mesomorphic samples does not break even if the strain reaches 1200%, indicating that these samples are ductile and flexible with greatly enhanced deformability [30,45].

Young’s modulus, yield strength and strain at break of all the samples estimated from the stress-strain curves are shown in Figure 11. It can be seen that, Young’s modulus and yield strength of composited γ-iPP samples are significantly higher than that of the neat γ-iPP, and the addition of 1 wt.% MWCNTs into γ-iPP can increase 20% in modulus and 28% in strength. The reinforcement of MWCNTs to the mechanical properties should attribute to their nucleation ability for crystallization of γ-iPP, which can improve the iPP-nanotube interaction [46]. Furthermore, because of the weaker crystalline texture or immature lamellar structure of γ-iPP, the strain at break of γ-iPP/MWCNTs is around 800% and increases to about 1420% higher than α-iPP/MWCNTs. In short, the γ-iPP/MWCNTs composites prepared by slow pressurization displayed more excellent mechanical properties than not only neat γ-iPP but also iPP samples in other phases.

For meso-iPP/MWCNTs composites, the Young’s modulus and yield strength are a little lower than that of neat meso-iPP, which is probably caused by two reasons. First, as discussed above, the formation of mesophase during rapid pressurization is through homogeneous nucleation, during which the MWCNTs barely have nucleating ability and just acts as the filler distributed in the mesophase matrix. When subjected to an external force, the local stress concentration around the MWCNTs can easily cause the matrix yield and rupture due to the weak interaction between mesophase matrix and the nanotube [44]. Second, the average size of the nodular domains of meso-iPP/MWCNTs are smaller than that of neat meso-iPP as given by SAXS, thus the meso-iPP/MWCNTs with thinner mesophase layer should show lower yield strength according to Young’s theoretical model of yield behavior [11].

## 4. Conclusions

In this work, we investigated the pressurization-induced crystallization behavior of iPP/MWCNTs composites within a wide range of pressurization rates from 2.5 to 1.3 × 10^4^ MPa/s. Two critical pressurization rates marked as **R_1_**, and **R_2_** were distinctly determined. If the pressurization rate is below **R_1_**, only γ-phase forms, and it can neither influence the crystallinity nor the lamellar thickness. When the pressurization rate is higher than **R_1_** but lower than **R_2_**, mesomorphic iPP begins to generate and coexist with γ-phase, meanwhile, for γ-phase the pressurization rate has a greater influence on its crystallinity than on its lamellar thickness. Once the pressurization rate exceeds **R_2_**, only mesophase forms due to the kinetic reason.

When iPP/MWCNTs composites solidified to mesophase, MWCNTs can increase the energy barrier for transport of molecular chains and result in the formation of nodular domains with smaller average size than that of neat meso-iPP. However, both the existence of MWCNTs and the nodular size hardly have impacts on the cold crystallization of meso-iPP/MWCNTs composites. In case that iPP/MWCNTs composites crystallized to γ-phase, the existence of MWCNTs can promote nucleation of γ-crystal and improve the iPP-nanotube interaction, and then successfully reinforce the modulus and yield strength which are comparable with that of α-iPP/MWCNTs, meanwhile, γ-iPP/MWCNTs composites also have high elongation at break, exhibiting the best overall mechanical properties among the prepared samples. Actually, this work provides an efficient method to tailor the crystalline structure and thus properties of iPP composites.

## Figures and Tables

**Figure 1 polymers-11-01294-f001:**
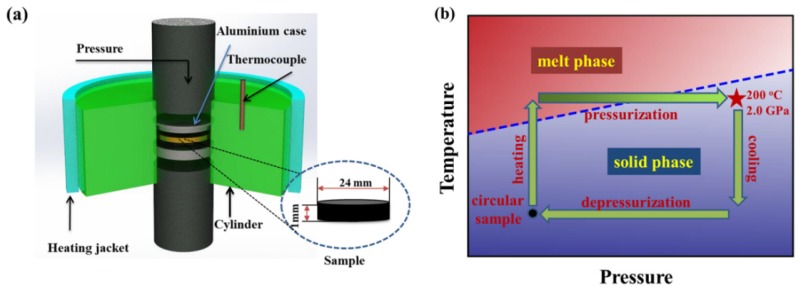
(**a**) Schematic of the high-pressure cell; (**b**) preparation diagram of pressurization treatment.

**Figure 2 polymers-11-01294-f002:**
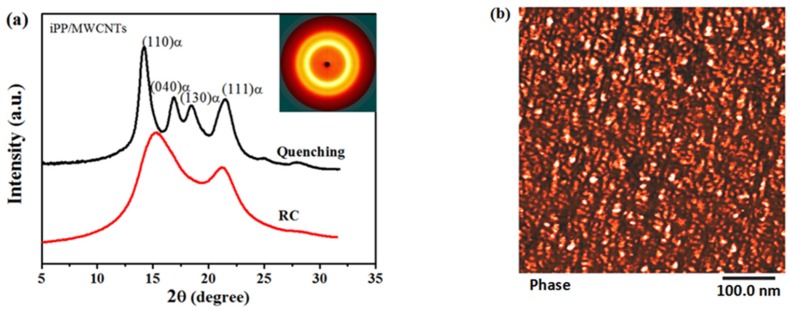
(**a**) Selected 1D WAXD profiles of the RC and quenching composites with the inset of 2D-WAXD pattern of RC sample; (**b**) AFM image of RC samples.

**Figure 3 polymers-11-01294-f003:**
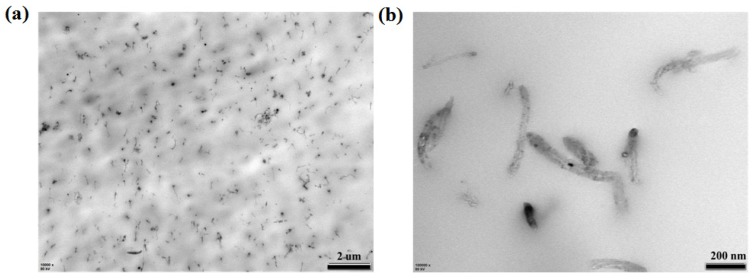
Transmission electron microscopy (TEM) images of RC samples at different magnification.

**Figure 4 polymers-11-01294-f004:**
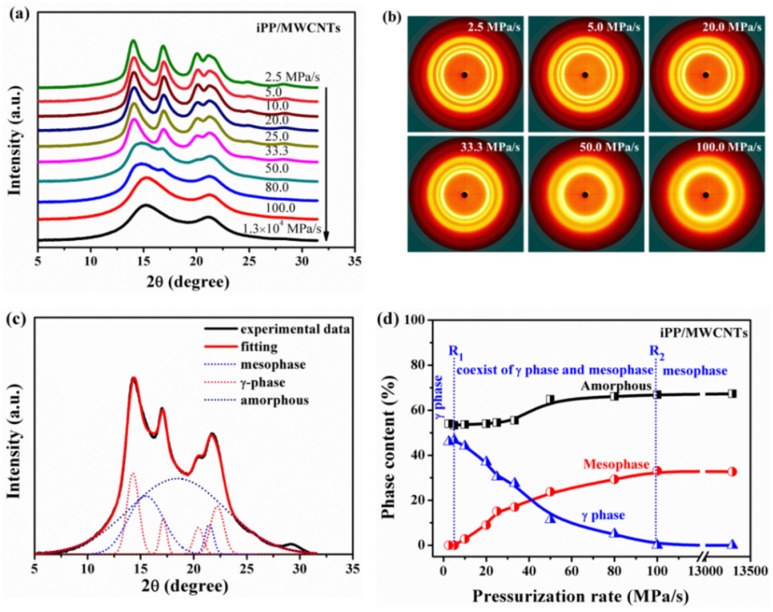
(**a**) Selected 1D WAXD profiles and (**b**) 2D WAXD patterns of multi-walled carbon nanotubes (iPP/MWCNTs) samples pressurized to 2.0 GPa at different pressurization rates; (**c**) the fitting curves of WAXD of amorphous, mesomorphic, and γ-phases; (**d**) contents of amorphous, mesomorphic, and γ-phases as a function of pressurization rate.

**Figure 5 polymers-11-01294-f005:**
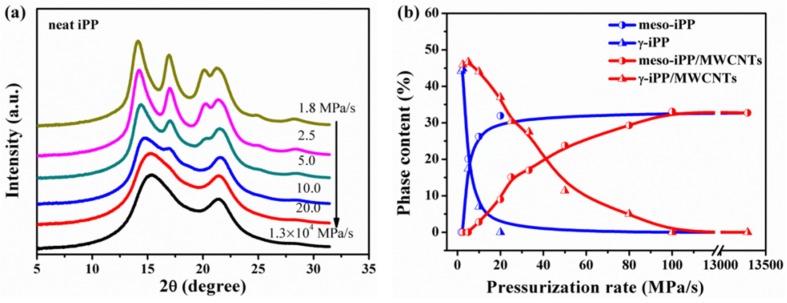
(**a**) Selected 1D WAXD profiles of iPP samples pressurized to 2.0 GPa at different pressurization rates; (**b**) contents of mesomorphic, and γ-phases as a function of pressurization rate.

**Figure 6 polymers-11-01294-f006:**
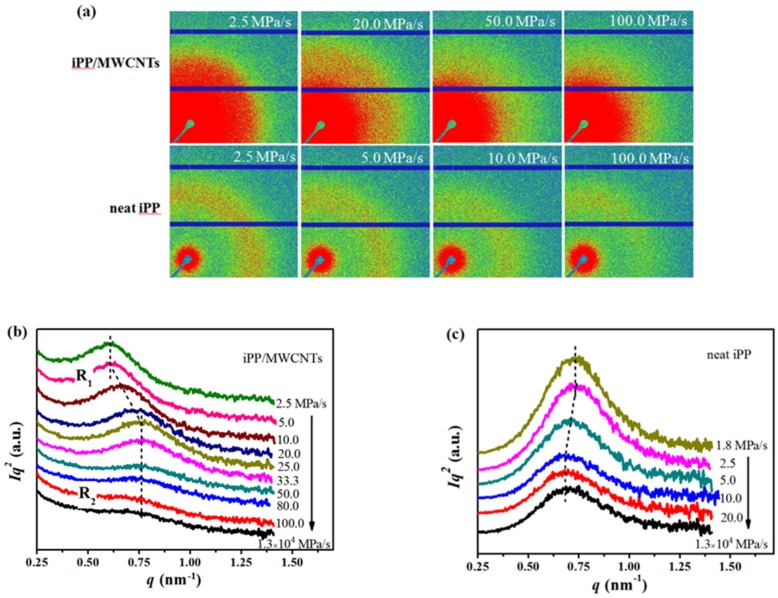
(**a**) Selected 2D small-angle X-ray scattering measurements (SAXS) patterns; 1D SAXS profiles of (**b**) iPP/MWCNTs and (**c**) neat iPP samples obtained by pressurizing the melts to 2.0 GPa at different pressurization rate.

**Figure 7 polymers-11-01294-f007:**
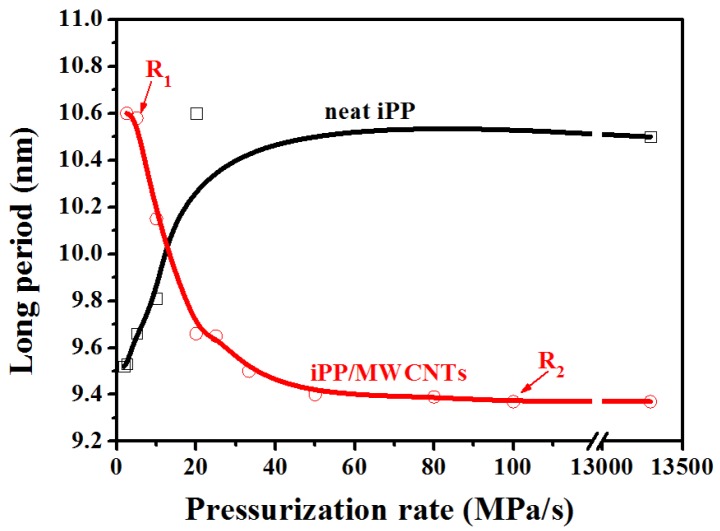
The long period of iPP/MWCNTs and neat iPP samples as a function of pressurization rate.

**Figure 8 polymers-11-01294-f008:**
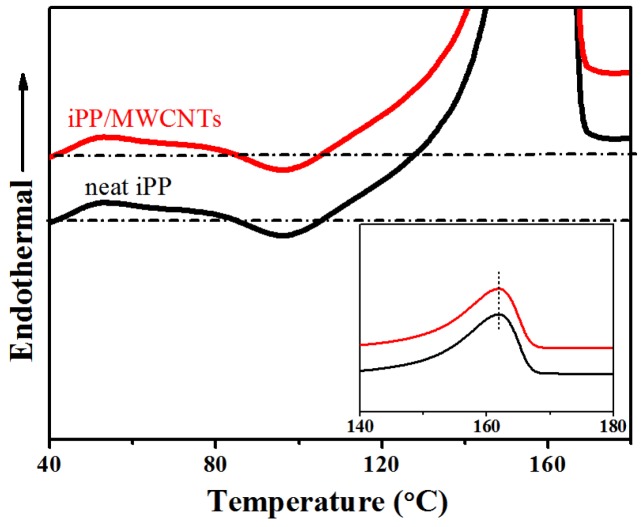
Differential scanning calorimeter (DSC) heating curves of the two kinds mesomorphic samples with the inset of the melting peak region.

**Figure 9 polymers-11-01294-f009:**
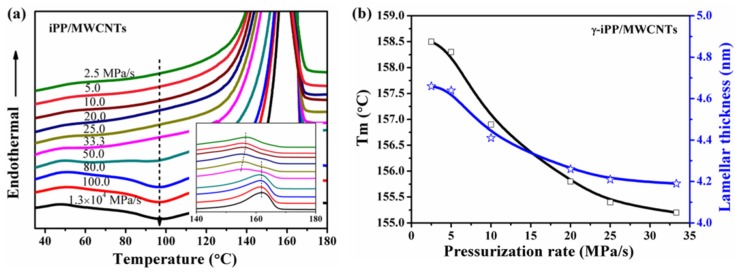
(**a**) DSC heating curves of iPP/MWCNTs samples obtained by pressurizing the melt to 2.0 GPa at different pressurization rate (the inset shows the melting peak region); (**b**) changes of $T_{m}$ and l of γ-iPP/MWCNTs samples as a function of pressurization rate.

**Figure 10 polymers-11-01294-f010:**
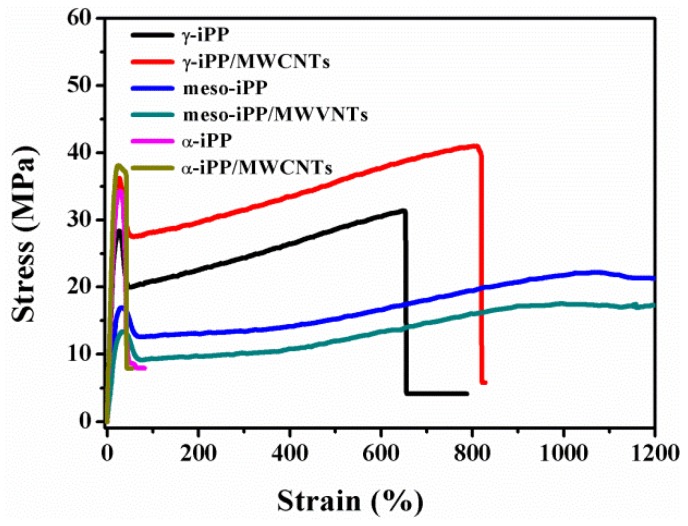
The typical stress-strain curves for iPP and iPP/MWCNTs composites.

**Figure 11 polymers-11-01294-f011:**
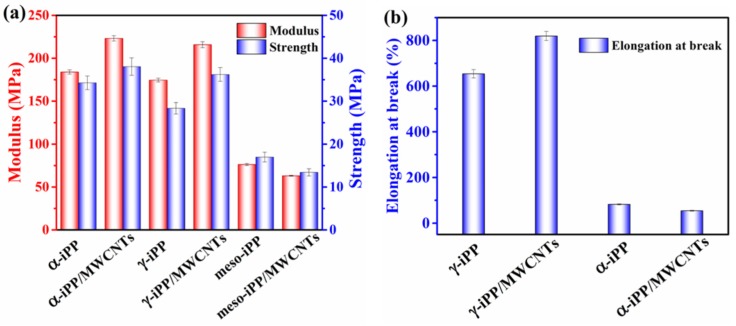
Mechanical properties about (**a**) modulus, yield strength and (**b**) elongation at break for neat iPP and iPP/MWCNTs composites.

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
