# Peer review of "Structure and Mechanical Properties of Multi-Walled Carbon Nanotubes-Filled Isotactic Polypropylene Composites Treated by Pressurization at Different Rates"

_polymers, 2019, doi:10.3390/polym11081294_

Round 1

Reviewer 1 Report

The paper is well written, and the topic is interesting. Authors have carried out a very good characterization of the nanocomposites by means WAXD/SAXD, DSC, TEM and AFM. The pressurization method is interesting to tailor the crystalline structure and properties of iPP/MWCNTs

Only some elucidations must be corrected:

Lines 32-33 and 403-404 can be consistent. In the abstract authors say that the pressurization method is proved to be an efficient approach…. Whereas in the conclusion section they say that maybe this work provides….

Line 75. Please, indicate exactly what you mean with superior

Line 131. Please, check if the detector Pila-tus is well written. I think that is Pilatus

Line 276.  c) is lost in the figure legend

Line 294. R1 and R2 are not labeled in the Figure for neat PP

Line 307. Monomorphic instead Mesomorhpic

Line 322. Exothermic instead Exthermic

Line 362. Figure 10. The X-axis should be extended to see the break of the monomorphic samples which occurs at 1200% of strain

Reviewer 2 Report

The manuscript itself is interesting and obtained results are bringing the advancement into the studied subject.

However, there are some points decreasing the overall quality of the manuscript:

part 2.3. - equations are not numbered.

Country of origin of the used instruments has to be given.

Fig. 11.: letters are too small to read them

list of References: left alignment is not present.

part 3.3.: equation is not numbered

Authors should explain, why exactly iPP of the grade T30S has been used. PP of this kind (with isotactic index of at least 95 %) is intended for raffia applications. Especially for ropes, geotextiles, carpet backings, woven bags and products of that kind are made of T30S iPP.  I think, authors just used anything available in the lab without further reasoning. I can imagine numerous PP grades to be of much better choice. By the way, PP in general is not the best polymer material for this study. Authors have to think about in next studies. Appropriate polymer substrate and its grade selection is of a paramount importance.

Round 2

Reviewer 2 Report

The Authors have provided satisfactory response to my comments.